# Using Mixed Methods to Understand Spatio-Cultural Process in the Informal Settlements: Case Studies from Islamabad, Pakistan

**Ramisa Shafqat \*** and **Dora Marinova**

Curtin University Sustainability Policy (CUSP) Institute, Curtin University, Perth, WA 6845, Australia
\* Correspondence: ramisa.malik@gmail.com

**Abstract:** A mixed-methods approach is used to understand the human factors defining cultural heritage in two informal settlements in Islamabad, Pakistan, namely France Colony and Mehr Abadi. The methodology applied is based on spatial investigation within a placemaking framework to create a visual representation of the neighborhoods, and grounded theory to explore the experiences and memories of their inhabitants through verbal communication. A combination of techniques, including transect walks, photography, and on-site interviews, allows us to map the tangible and intangible elements of the informal settlements. Cultural characteristics are identified as essential in the spatio-cultural processes occurring in the informal settlements. The study concludes that cultural dilapidation happens because of obstructions in the processes translating intangible heritage into tangible space. Appropriate policy interventions are suggested to minimize the loss of rural heritage transfer to informal settlements within the urban fabric of Islamabad.

**Keywords:** mixed research methods; urban informal settlements; spatial analysis; social analysis; Islamabad; sustainability

## 1. Introduction

While urbanization in the Global South is largely considered to be synonymous with progress and development, its benefits often fail to cover socio-cultural gaps existing within a city by ignoring or circumventing the reality of informal settlements. Informal settlements or informalities are often seen as "a heavily populated urban area characterized by substandard housing and squalor" [1] (p. 8) or are characterized as places "of terrible housing, foul drainage and inadequate sewerage, abundance of bugs and dirt, extreme un-healthiness and population of transients, criminals and the unskilled living in extremely insecure and impoverished circumstances" [2] (p. 1). The contribution of informal settlements in weaving the urban fabric is regularly disregarded, and their population is consistently marginalized spatially and socially [3]. Under such a view about informalities, policies are proposed which aim at either formalization, regularization, or eradication of these settlements [4–7].

More recently, initiatives have been put in place to collect and make available data on internal migration that drives the growth in urbanization [8]. Attempts have also been made to geographically map the growth of informalities across the planet, which account for a third of the total urban form and where it is expected that up to two billion people will live by mid 21st century [9]. This helps understand the scale of the problem and the unique morphology of the urban informalities; however, the socio-cultural contribution they make continues to be difficult to capture.

Pakistan is a good example where informal settlements are rarely perceived or appreciated from a socio-cultural perspective. Their contribution to the urban fabric and its sustainability is commonly ignored. Urban policies remain indifferent to the socio-cultural dimensions of a settlement, and regulations regarding housing of poor people,

rehabilitation of suburban areas or upgradation of living quarters revolve around tenure status, infrastructure, amenities or demographic characteristics, while the soft assets of the informal settlements are often not included in the planning process. This study is based on the hypothesis that informal settlements have a socio-cultural potential as reservoirs of cultural heritage due to assimilation of rural migrants from the surrounding region. This phenomenon is further accentuated in a modern planned city, such as Islamabad built from scratch to serve as the capital of Pakistan.

The significance of the rural culture that the dwellers of the informalities bring to the city is far from being only a basic socio-economic function and agrarian lifestyle. It is imbued with rural wisdom that has travelled through centuries to reach the present time [10]. Pakistan represents an interesting canvas to understand the elements of rurality found in the country's urban informal settlements. Traditionally, urban centers in Pakistan have grown around historic fortifications [11] with rural migration contributing to their expansion. Islamabad, however, displays a new form of urbanization created in the 1960s through futuristic design and planning with inputs from some of the best-known western and eastern planners (i.e., Constantinos Doxiadis) and architects (i.e., Le Corbusier, Gio Ponti, Robert Mathews, Walter Gropius, Kenzo Tange, Edward Durell Stones, Arne Emil Jacobson, and Louis Isadore Khan) from the developed world [11]. Islamabad's Capital Development Authority (CDA)—a public agency responsible for the delivery of services in Pakistan's capital, also has a team of planners and architects who shape the development of the city. Rural immigrants residing in Islamabad's informal settlements (locally called Katchi Abadis) play a significant role in creating densely populated areas with a substandard level of housing, and also in introducing rural cultural heritage into the capital's contemporary urbanism. The originally designed idyllic picture of the capital city has been significantly modified by organic growth, which creates a cultural clash between the administrative elite living in the formal areas and the urban migrants inhabiting the informalities [12]. Islamabad has burgeoned organically and informally, in defiance of projected plans [13]. It currently has more than 50 informal settlements, 11 of which are recognized by CDA and consequently not subject to evictions [7], while the others are considered only as a temporary phenomenon. Two informal settlements—France Colony and Mehr Abadi—the former recognized by CDA and the latter not, are investigated in this study using a mixed-methods approach.

This paper methodically examines the characteristics of informal settlements and the socio-cultural traits of their residents, advocating for their socio-spatial recognition from a "rights to the city" perspective [14]. Using a mixed-methods approach, it identifies these socio-cultural traits to be of immense value due to their sustainability virtues. By exploring the spatial as well as socio-cultural aspects of the informal settlements, the aim of the study is to comprehend the elements of cultural heritage that entail sustainability elements and synthesize them for a better understanding of the contribution of these settlements in order to make cities more sustainable. In the remainder of the paper, we first explore the link between sustainability and informal settlements, as outlined in high-level policy documents. We then explain the two-pronged theoretical approach adopted in the study of two informal settlements in Islamabad, which builds on a spatial analysis and grounded theory. The types of methods used under each of the two frameworks are also explained. This is followed by the specific application of the mixed-methods methodology to the two case studies and generation of analytical insights that argue the importance of rural spatio-cultural processes in contributing towards the sustainability of the informalities and the overall urban settlement. Policy recommendations are also outlined in order to bring together values, community, architecture, nature and economic activities in a viable relationship between culture and space.

This paper expands on the previous work by the authors which separately analyzed placemaking [15] and grounded theory [16] in relation to the Pakistani informalities. The approach adopted in the current paper provides an overarching holistic picture of the informalities and focuses on the two-pronged methodological framework, in which grounded

theory and spatial (placemaking) analysis sit complementing each other. Hence, the aim of the paper is to highlight how the findings from each wing of the two-pronged methodology overlap and inform each other to consolidate and validate the data collected from the mixed-methods approach.

## 2. Sustainability and the Urban Agenda

The United Nations (UN) Habitat Conference on Human Settlements held in Vancouver, Canada in 1976 shifted the world's attitude towards informal settlements for the first time. It was stated in subsequent UN documents that: "It was the first time that governments universally acknowledged that slum and squatter settlements could play a significant role in the national development process" [17] (p. 5). Informal settlements were no longer described as an isolated or temporary phenomenon, but rather as part of the urbanization process.

Improving housing conditions has been a long-lasting concern of the United Nations (e.g., United Nations [18]). The United Nations Sustainable Development Goals (SDGs) adopted in 2015, and in particular SDG11 on making cities and communities sustainable, emphasize inclusive and sustainable urbanization [19]. In 2016, the New Urban Agenda was adopted at the United Nations Conference on Housing and Sustainable Urban Development Habitat III in Quito, Ecuador [20]. It laid down principles related to planning, development, implementation, and management within a participatory framework based on creating inclusive and resilient urban communities, including in the informal settlements.

The New Urban Agenda acknowledges that culture and cultural diversity are sources of enrichment for humankind, and that they provide an important contribution to the sustainable development of cities, human settlements and citizens, empowering them to play an active and unique role in development initiatives. It further recognizes that culture should be taken into account in the promotion and implementation of new sustainable consumption and production patterns that contribute to the responsible use of resources and address the adverse impacts of climate change [21] (p. 4). Cultural diversity is acknowledged as an attribute of sustainable communities within the New Urban Agenda, as well as in the SDGs.

Potential remnants of rural cultural heritage in urban informalities are components of the fabrics of cities and should be part of their quest for sustainability. However, how do we identify and start to understand the spatio-cultural contribution of informal settlements? High-level policy documents alone cannot deliver the knowledge needed on the ground to formulate policies and planning approaches that do justice to the cultural heritage and diversity manifested in the urban life of the informalities. We explore this by taking as examples two informal settlements in Islamabad, Pakistan, and we apply a mixed-methods approach to investigate their spatial and socio-cultural realms.

## 3. Methodological Framework

The study's objective is to investigate tangible as well as intangible dimensions of cultural heritage which is transported into the informal realms of the city either physically or through memories, experiences and practices associated with the past rural lifestyles of the migrants who have settled in the urban informalities of Islamabad. Due to the nature of the topic, which tries to capture complex aspects of urban life, a mixed-methods approach is adopted to generate insights aimed at contributing to improved understanding and better planning. Mixed methods have been used in the last 30 years or so [22]. Their application allows the problem to be studied from different perspectives [23], and helps to provide a deeper meaning and more complete vision of the studied phenomenon compared to only one method of investigation [24]. Another distinctive feature of mixed-methods research is that it focusses on practical advice [22] rather than particular philosophical stances or statistical descriptions. These features make this approach very well-suited for analyzing urban informalities with the aim to understand the spatio-cultural processes that occur in them. Despite the overwhelming support for mixed-methods research, this

approach has some drawbacks, as it requires the researcher to be familiar with a range of methodologies, and it is generally more time consuming. Furthermore, it may produce conflicting data and results which the researcher has to interpret in order to understand the studied phenomenon [25]. In this particular study, we did not face the latter problem and were able to generate good practical understanding of the urban informalities.

The mixed-methods approach builds on two frameworks which are respectively adapted for visual and verbal data collection, namely the spatial framework and grounded theory. We refer to this as a two-pronged approach (see Figure 1) with the aim to collect socio-spatial data to fully investigate the dimensions of the informal settlements, their attributes and the stakeholders involved as drivers of change. The visual data aid in collecting spatial information and the verbal data aim at gathering spatio-cultural information.

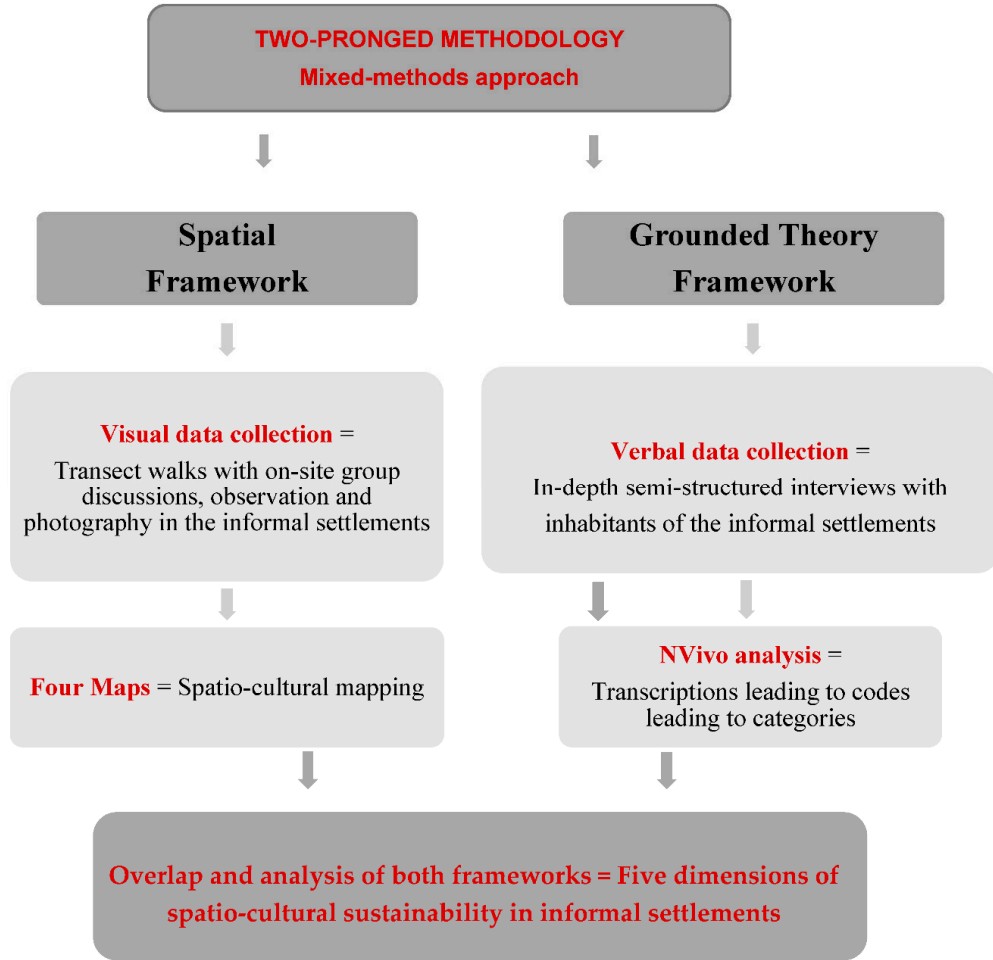

**Figure 1.** Research methodology.

Within the spatial framework, a suite of different methods was used to conduct a thorough analysis of the two selected sites. It included transect walks around both sites with on-site group discussions, observations and photography. This allowed for detailed maps to be created of the informal communities to facilitate appropriate data analysis [26–28]. Grounded theory was used in this data analysis to extract socio-cultural dimensions of the informal settlements that have been overlooked in the current theoretical discourse and policy in Pakistan. Based on collecting primary verbal data through in-depth interviews with inhabitants of the two sites, the aim was to develop a unique theoretical perspective that reflects their cultural heritage and represents the informal settlements [29,30]. Further detail about the use of these two frameworks is presented in the subsequent sections below.

*3.1. Spatial Framework*

As place is created when space comprised of spatial dimensions is humanized by socio-cultural elements [31–33], it is, hence, important to have a way to capture the spatio-cultural aspects of the informal settlements. A lot of work by urban practitioners has been undertaken to understand, explore and document places and their characteristics. The place diagram (see Figure 2) developed by the Project for Public Spaces (PPS), a US-based cross-disciplinary not-for-profit organization established in 1975 [34], is used as the foundation for the visual data collection, which then allows for maps to be created. This diagram is refined and contextualized for the informal settlements of the Global South and represents the spatial framework for the investigation to ensure that no important characteristic is omitted.

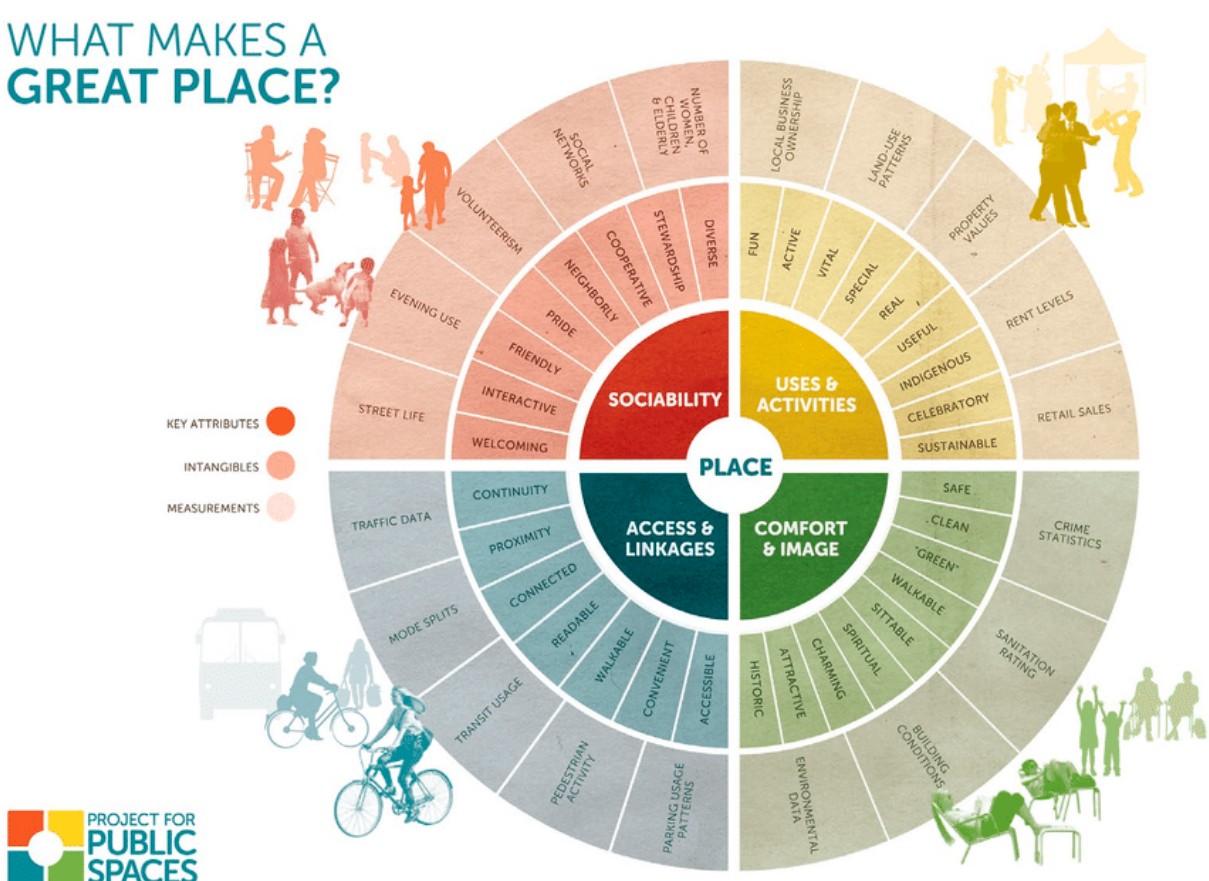

**Figure 2.** Place diagram developed by Project for Public Spaces (PPS). Note—Project for Public Spaces, n.d. (https://www.pps.org/article/grplacefeat accessed on 2 May 2020). Copyright—granted.

For PPS, "placemaking facilitates creative patterns of use, paying particular attention to the physical, cultural, and social identities that define a place and support its ongoing evolution" (https://www.pps.org/article/what-is-placemaking, accessed on 2 May 2020). This conceptual ideology supports placemaking as a process involving the socio-cultural elements of space. The basic categories of "place" defined by PPS, namely uses and activities, comfort and image, access and linkages and sociability (see Figure 2), are included in the observations, group discussions and on-site interviews during the transect walks in the informal settlements. These categories are subsequently re-defined to better represent the informal settlements. The PPS model of place definition is used for data collection purposes only with the intention to translate this information into maps of the informal settlements. Four maps are prepared for both case studies, informed by the PPS diagram, which visualize the four attributes adapted to the informal settlements. The two specific methods used are transect walks and mapping.

3.1.1. Research Method: Transect Walk

This method allows researchers to explore the socio-cultural dimensions of a particular place [35]. It requires members of the community, accompanied by a researcher and other technical people—in the present study these were two architects—to take a walk through the neighborhood while having interactive discussions, recording key elements and gathering visual documentation by means of instant sketches, photography and discussions with locals passing by and with other users of the communal spaces [36]. The method of the transect walk aims at investigating the "articulated moments in networks of social relations and understandings" [37] (p. 154) within the informal community.

Several different routes were taken for transect walks during each visit to the sites to fully understand the place and its important characteristics. This method makes it easy to have discussions with community members about specific locations and gives a good perspective about their routine needs, local behaviors, everyday challenges and space constraints. The transect walk helps in conducting on-the-spot discussions about the particular place (see Figure 3), with instant feedback gathered from community experiences (Catalytic Community, http://catcomm.org, accessed on 2 January 2019).

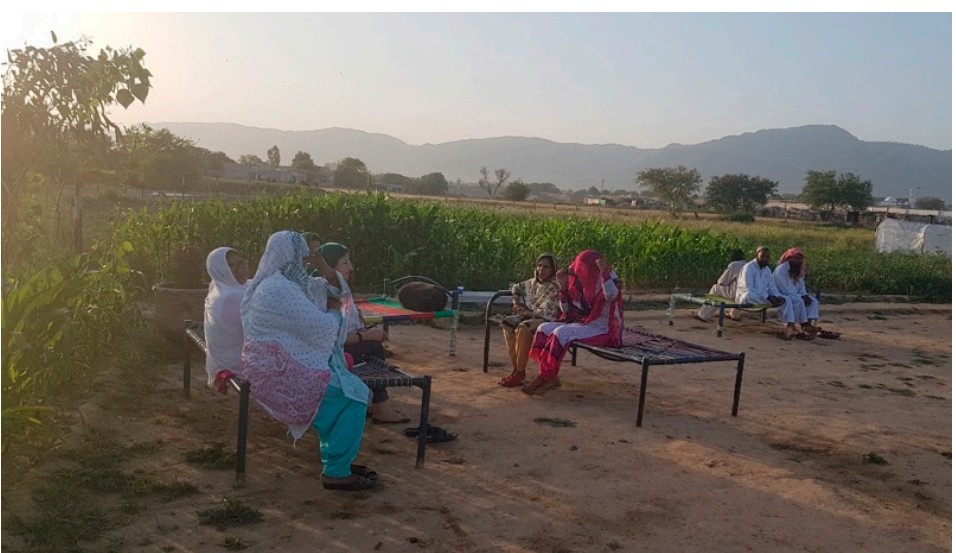

**Figure 3.** Group discussion during a transect walk at Mehr Abadi. Source—author.

Using observation, on-site discussions and photography, the transect walks investigate a range of different elements, as follows:

- Local architecture and streetscape;
- Public transit access points and pedestrian desired routes;
- Public activity and dynamic use of space by different users;
- Natural and topographic constraints;
- Vacant spaces/potential areas for future development;
- Street commerce and micro-economy;
- Common community areas and amenities;
- Unique characteristics of public spaces and symbols;
- Sanitation and contaminated spaces;
- Expression of art or ideas, including graffiti and murals.

The participants were engaged in informal discussion about these elements during the walk, after having initially brainstormed their place appreciation before setting out. Open-mindedness in recording expected and unanticipated data were also maintained. Photographs were taken of particular features of the informal settlements.

### 3.1.2. Research Method: Mapping

Socio-cultural maps were prepared from the information gathered during the transect walks for further analysis of the informal neighborhoods' spaces and to explore potential placemaking interventions. The aim of these maps is to record the flow of people and invisible elements of a place, particularly those which other visual data collection methods, such as photography, cannot capture. These maps depict the data gathered during the transect walks along with the physical elements of the place. The maps were prepared with the help of geographical information system (GIS) software, i.e., ArcGIS, Google Earth, AutoCAD and Photoshop software, and supported with on-site photography.

The transect walks covering site visits, architectural observations, and questions asked during the interviews helped to create the four maps (see Figure 4) which was deemed an essential task as it provided a medium to depict the empirical data of the spatial characteristics of the settlements and pinpointed social activities in terms of spatial configuration. This exercise consolidated the socio-spatial data collected during the transect walks and gave a clear understanding of the place documented on paper. As explained by Corner [38] (p. 213), "[m]apping ([is] a collective enabling enterprise, a project that both reveals and realizes hidden potential", therefore, "creating and building the world as much as measuring and describing it".

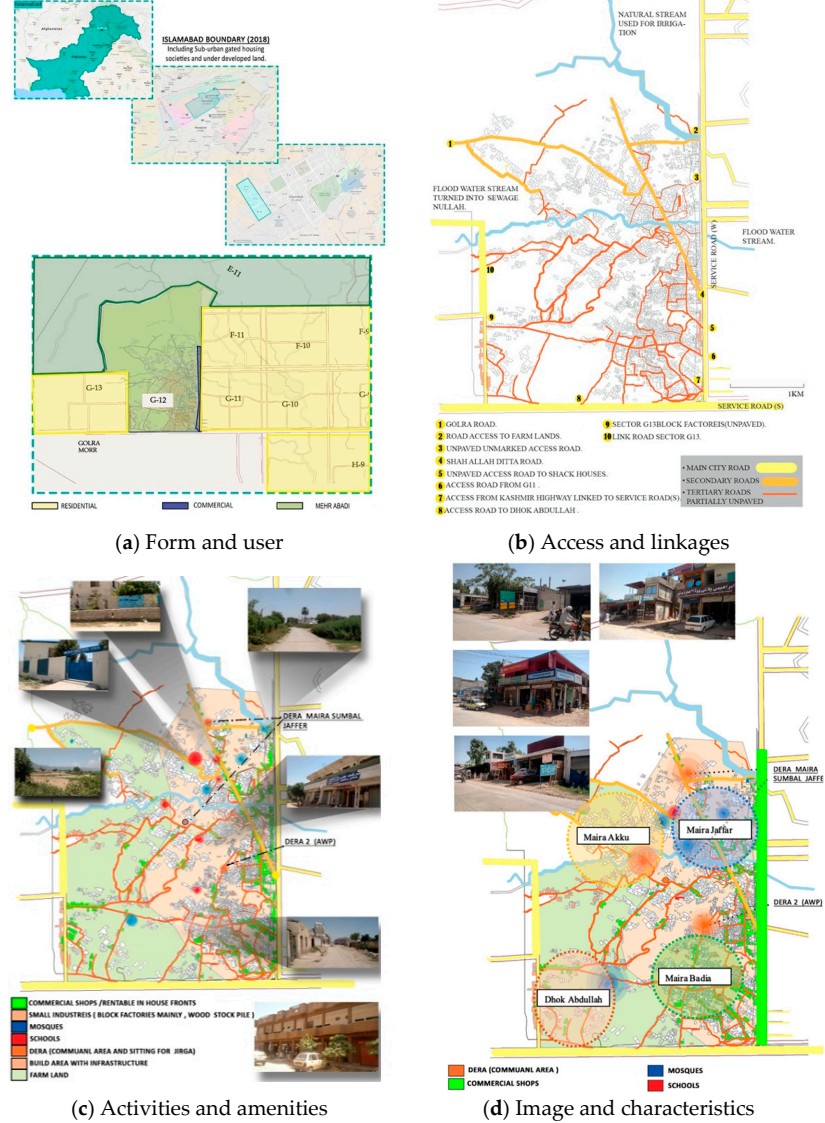

(**a**) Form and user

(**b**) Access and linkages

(**c**) Activities and amenities

(**d**) Image and characteristics

**Figure 4.** Four maps of Mehr Abadi—an example.

Raw data in the form of photography, mental maps and scribbles are also used to provide information for the map creation. Pictures are usually categorized and used as tools of visual ethnography. The photographs taken documented the typology of houses, streets, bridges and other built environment characteristics in the field sites and helped in deciphering the meaning of the place. Other visually recorded elements include rural imagery, adults' and children's activities, animals and other communal happenings in the public space.

*3.2. Grounded Theory Framework*

Grounded theory is a concept laid down by Glaser and Strauss [39], which transcended traditional qualitative research theory from its descriptive nature into the formation of a theoretical framework constructed from abstract explanations of social processes, underpinned by comparative analysis with an open mind [40]. It is used to produce a theoretical framework to study an area that is unexplored, assuming that " . . . all of the concepts pertaining to a given phenomenon have not yet been identified, at least not in this population and place. Or, if so, the relationships between the concepts are poorly understood or conceptually undeveloped" [30] (p. 53). The existing gap in assessing various phenomena associated with informal settlements in Pakistan from a culture and sustainability perspective has stimulated the use of grounded theory in this research. It is expected to reveal undiscovered and latent dimensions of urban regeneration processes in the marginalized communities of Pakistan.

The analytical process at the core of the grounded theory methodology is coding as a technique of labelling and categorizing transcripts to extract deeper meanings from the lines of information and to find connections between different categories. Coding needs to be carried out effectively and meticulously [40] to allow for new meaning and understanding to emerge. For this study, NVivo software was used to analyze the codes and create categories which explain the characteristics of the informal settlements.

Two specific methods used within the grounded theory framework are memo writing and in-depth interviews. The empirical data are collected following the norms of a theory building strategy, with the inclusion of placemaking socio-cultural analysis. Thus, the grounded theory method assimilates the findings from the placemaking analysis, which leads to the theory creation process.

### 3.2.1. Research Method: Memo Writing

Memo writing is an essential part of the grounded theory methodology. It is a way for the researcher to record field notes, ideas, feelings and experiences during site visits, and visual reactions of the informants, in a systematic and chronological manner [39]. As the research is investigating the social elements of space, memo writing played an important role to keep a record of intangible elements experienced within the site's spatial boundaries. As it is very fruitful in facilitating the understanding of the social and cultural experiences encountered during site visits, memo writing helps the researcher to take these aspects into account while performing an analysis of the data collected [40]. The written memos were effectively used to capture the characteristics of both case studies.

### 3.2.2. Research Method: In-Depth Interview

An in-depth interview is a qualitative research method that enables a direct interactive conversation with the concerned person and gives an insight into their thought processes, behaviors, and emotional attachments with the issue at hand [41]. According to Strauss and Corbin [30], in-depth interviews are an important tool for investigating and extracting valuable data within the grounded theory methodology.

In the present study, in-depth interviews are the main source for qualitative data collection about the socio-cultural dimensions of the informal community. In-depth interviews were carried out with dwellers of the informal settlements as a means to connect with their cultural heritage, ways of thinking and aspirations. In total, 17 interviews were conducted

for each of the case studies. The coding of the interviews further reveals the underlying values and symbols adopted by these people [41].

Figure 5 depicts the casual environment in which interviews were conducted. Interviewees from different gender, age, employment and social clusters were deliberately selected to cover a diverse range of the populace. However, the governing factor in sampling was the willingness of respondents and convenience of access. The interviews were conducted over a period of two months and multiple trips to the sites along with the community gatekeeper—an influential and trusted local resident who, through their existing networks, provided access to other people active in the neighborhoods within the settlement [16]. The majority of the interviews were in group settings with participants across different age groups. Both, women and men participated in these interviews, and occasionally children were also present. We were not looking for a demographic representation across a diverse cross-section of the population, e.g., according to age, gender, occupation or class status, as some of these characteristics are not apparent or redundant in a group setting. It was impossible to capture the overall cultural make-up of the informal settlements. Not all trips were successful, but since each visit and interview lasted approximately 40–60 min, it was feasible to visit as per the preference of the residents and availability of the gatekeeper.

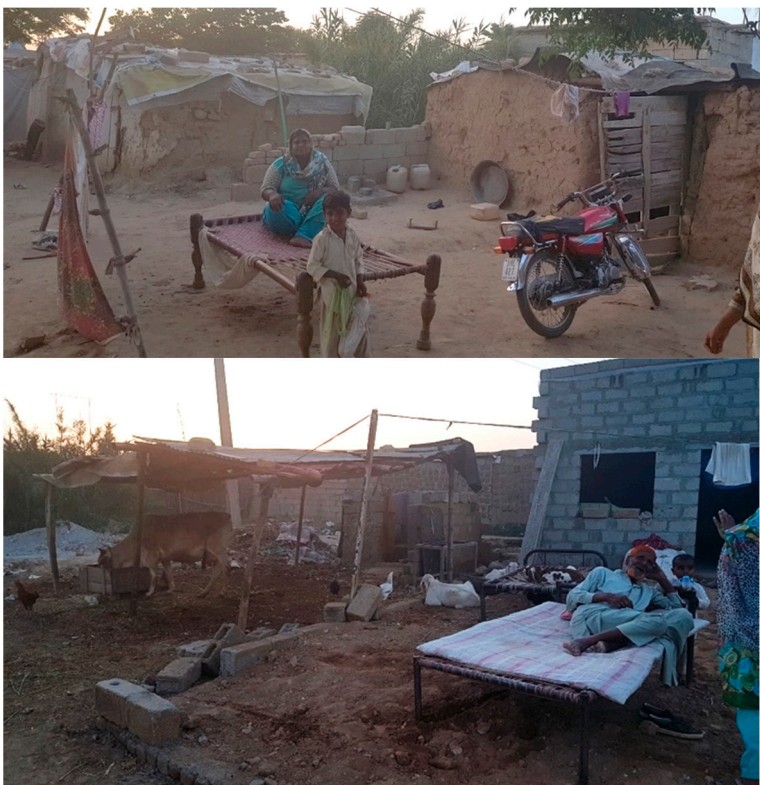

**Figure 5.** Interviewees in their setting.

## 4. Application of the Mixed-Methods Approach

This section consolidates the findings from the two-pronged methodology, namely the spatial framework (based on visual findings) and grounded theory (based on verbal findings). Their overlap and analysis generate five dimensions of spatio-cultural sustainability manifested in the two case studies. The use of the mixed-methods methodology allowed us to decipher the spatio-cultural process in the informal settlements.

### 4.1. Spatial Framework Findings

The elements deduced from the four maps (see the example in Figure 4) built a pragmatic contribution to the theory generated and helped eliminate some assumptions

made in the findings from the verbal data collection about spatial contexts. In fact, the visual data analysis provided a reality check to ensure theory formulation that is close to the context and can be effectively used for local policy and urban practices in Islamabad. Table 1 shows the summary of findings from the spatio-cultural maps. The text in *italics* highlights constraints/barriers in cultural proliferation identified during the data collection within the spatial framework.

**Table 1.** Spatial process in informal settlements.

| France Colony | | Mehr Abadi | |
|---|---|---|---|
| **Form and user** | High density | **Form and user** | Medium density |
| | Interconnected built environment | | Cluster of settlements |
| | Metro-core | | Peri-urban |
| | Christian majority population | | Muslim majority population |
| | Regularized tenure | | *Unregularized tenure* |
| **Access and linkages** | Organic pattern of road network | **Access and linkages** | Organic pattern of road network |
| | *Non-permeable periphery* | | *Non-permeable periphery* |
| | Streets are dynamic spaces | | Streets are dynamic spaces |
| | Sustainable mode of transportation | | *Limited automobile access especially to farmland* |
| **Activities and amenities** | Traditional bazaar concept | **Activities and amenities** | Traditional bazaar concept |
| | Self-reliant island | | Connected with nearby markets, villages and peripheral settlements |
| | Mixed-use building configuration | | Self-made infrastructure and amenities |
| | Micro-finance | | Factories and cottage industry |
| | Multi-purpose use of common open spaces | | Farmland—an opportunity for urban farming |
| | Significance of churches and Dera | | Significance of Dera and mosques |
| **Image and Characteristics** | Spill-over spaces | **Image and Characteristics** | *Unwelcoming, commercial, and non-legible formal–informal interface* |
| | Multiple users of public place | | Cultural diversity and limited interaction amongst clusters of settlement |
| | Vernacular architecture | | Diverse housing typology, as follows: (a) block construction, plastered walls and concrete roofs; (b) portable roof or temporary roof structures; and (c) squatter houses or shacks |
| | Art expression, attire and lifestyle | | Rural imagery |
| | Crime, drugs and image of the settlement | | Public places and festivals |
| | *Pollution and sanitation issues especially Naalah serving as landfill site* | | *Garbage dumped on vacant land between cluster of settlements and Naalah* |

Here, Naalah means a fresh water channel, a Dera is a communal meeting place, and a bazaar is a market place.

A range of significant sustainability concepts present and practiced by the community were brought to the fore either intentionally or accidentally. They include self-reliance, mixed-use and incremental architecture, urban farming, communal cohesion, and self-governance and cultural showcasing.

The analysis shows that sustainability attributes reflected in the cultural values of the rural migrants in the informal settlements transcend the modern industrial and urban trends as a paradigm shift from conventional urban planning and development to organic placemaking. They are in fact deeply embedded in the cultural heritage of the region and are being practiced in the informal settlements of the cities without being acknowledged, documented or explored [15]. As one of the participants explains: "Here, there are a lot of people who are trying to keep their rural heritage alive and have not changed their

lifestyle". Another participant highlights his desire to continue the family tradition of supporting people in need, as follows: "I have made the front bedroom of my house a motel (sarayee), so that whoever migrates from the village can stay a few days here until he [sic] finds a place to live . . . my father used to host guests from other villages free of charge in our ancestorial village too, I wanted to keep his legacy alive". Rural pride and wisdom are also expressed in simple words rejecting modern-day consumerism, as follows: "our land is our mother; it produces everything from itself which we can thrive on" and "wealth escapes up in the sky like an eagle, because if money is not earned under good values, it is not sustained either".

The current government policy disregards the socio-cultural make-up of the particular informal community and its influence on its surroundings, its sustainability content, cultural elements of the settlement, social setting, and cultural impact of a certain informal community on the formal sector. Without being well-documented, the intangible elements of social and cultural heritage cannot be incorporated into the planning and policy making process. However, it is contended that the socio-spatial documentation of cultural heritage by applying the proposed spatio-cultural mapping methods in the urban planning process will result in policies which are more sensitive towards safeguarding and preserving cultural assets of the informal community. This equally applies to tangible and intangible heritage [42].

The current study respects the cultural variables of the informal community and investigated the spectrum of spatio-cultural aspects which standard explorations of informal settlements in Pakistan tend to disregard. Any meaningful policy making must take into account the soft assets contained within informal settlements. A pre-requisite to policy making would, thus, be a systematic documentation of such characteristics. Without careful socio-spatial mapping of the informal settlements, a successful policy cannot be devised, and there would be a significant risk of a loss of cultural heritage in an attempt to physically upgrade them.

### 4.2. Grounded Theory Findings

Grounded theory is used in the study to explore new perspectives and approaches in the unexplored areas of culture and sustainability in the informal settlements of Islamabad. The grounded theory findings are synthesized through the five stages of analysis suggested by Braun and Clarke [43], namely data familiarization, coding with NVivo, looking for themes, and scrutinizing and defining the themes. Four distinctive themes were deduced, namely (a) values and social practices—ancestral wisdom, simplicity, adaptability and self-reliance; (b) communal networks and relationships—social cohesion, informal justice system, tolerance and diversity, leadership, volunteerism, and governance; (c) built environment and ecology—self-built, incremental and vernacular architecture, inclusion of flora and fauna, pollution crisis, and indigenous waste management strategies, and (d) remnants of rurality—traditional craft and livelihood, traditional cuisines, folk games and festivities [16]. They are all linked to sustainability characteristics present in the informal settlements. It is also important to emphasize that within the context of this research, the participants understood culture and their role in the urban environment as culture carriers—people who continuously preserve or "carry" certain values, practices, concepts and behaviors originating from their rural roots [44]. Culture influences the decisions made by the dwellers of the informalities by providing rules and principles to guide their behavior and the choices they made while preserving cultural knowledge [45].

The grounded theory findings show that several popular sustainability concepts are in fact deeply embedded in the cultural heritage of the region and are slowly and gradually diminishing due to the advent of modernization, migration and urbanization in Pakistan. They include the concept of recycling, self-reliance, incremental housing, adaptability and culturally-based employment opportunities. Discovered during the in-depth interviews, these sustainability characteristics represent a corner stone of the indigenous lifestyle in the informal settlements.

*4.3. Understanding the Spatio-Cultural Process in Informal Settlements through a Mixed-Method Approach*

Using the two-pronged methodology, it became evident from the findings that the informal community struggles to preserve and practice cultural processes based on its rural values, memories, and experiences within the urban setting (see the italicized text in Table 1). This includes the following barriers/constraints:

- Pollution;
- Tenure insecurity;
- Poor access and transportation;
- Formal–informal divide;
- Poor infrastructure and sanitation.

The constraints hampering the process of cultural proliferation into spatial manifestation can limit the translation of cultural heritage into place, leading to loss of heritage and sustainability practices. Figure 6 describes how the intangible cultural heritage of the rural–urban migrants is expressed through the spatial construct of the informal settlements. The inhabitants of the informal settlements play the role of culture carriers, bringing intangible cultural heritage built around sustainability values from their ancestorial villages into the city. This heritage is translated into the spatial realm in the manifestation of informal settlements. However, where the translation of intangible culture into tangible spatial/physical fabric faces constraints, a loss of regional heritage and traditional practices supporting valuable sustainability concepts and practices occurs.

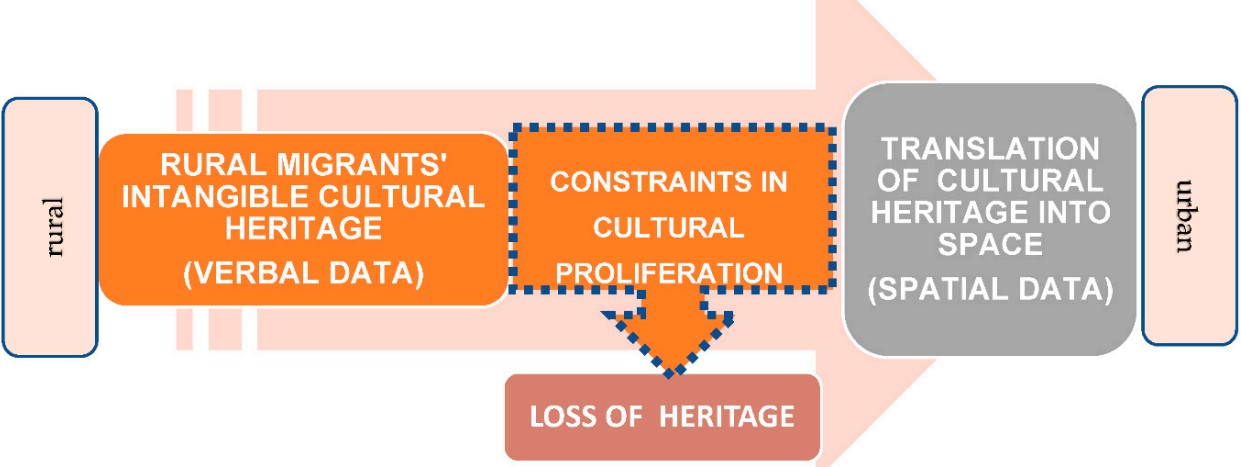

**Figure 6.** The role of rural migrants in the spatio-cultural process.

Moreover, it is contended that as the research was carried out by a mixed-methods approach, overlapping of results from both methodologies is key to a comprehensive analysis. Table 2 puts the spatio-cultural findings of the case studies (derived from visual data) against the categories which emerged from the grounded theory methodology (derived from verbal data). This helps to converge the findings through the two frameworks and investigate the relationship between them. Table 2 manifests the interlacing of the two frameworks, and the findings reveal that the two approaches have led to mostly similar categories. As a result, five main areas of sustainability interest immerged, namely informal settlements which are value-led, communal, with specific architectural features, oriented towards the natural environment, and livelihood-led. These categories derived from the overlapping of verbal and visual data are the spatio-cultural elements that need to be undertaken to preserve the tangible as well as intangible assets of the informal settlements.

**Table 2.** Consolidating findings from the mixed-methods approach.

| | Findings from Verbal Data ➤ | | | | Spatio-Cultural Elements | ◄ Findings from Visual Data | | | |
|---|---|---|---|---|---|---|---|---|---|
| **Values and social practices** | Ancestral Wisdom | Adaptability | Simplicity | Self-reliance | **Value-led** | Self-reliant Island | Expression of art, Attire and lifestyle | Multi-purpose use of common open spaces | |
| **Communal networks and relationships** | Social cohesion | Informal Social Justice | Tolerance and diversity | Leadership and governance | **Communal** | Dera (Place for communal activity) | Organic Pattern and cluster of communities | Role of church and mosque | Cultural diversity and Cluster of settlements: Populated by Baradaris of similar origin |
| **Built environment** | Architecture Self-built, | Recycling technique | Pollution crisis and indigenous waste | | **Architectural** | Vernacular architecture | Street as dynamic places | Self-built | Mixed-use building configuration |
| | | | | | **Natural** | Rural imagery | Farmlands- An opportunity for urban farming | Significance of Naalah and old Banyan trees | |
| **Remnants of rurality** | Traditional craft and livelihood | Traditional Cuisine | Folk games and festivals | | **Economic** | Traditional Bazaar with diverse shop typology | Factories and cottage industries | Public places and festivals | Micro-finance and en-trepreneurship |

A wider picture needs to be seen in order to understand the spatio-cultural process within the informal settlements in a way which is mindful of the social, cultural, economic and infrastructural setting of the inhabitants. Existing policies focus mainly on tangible elements, while the intangibles have been ignored. The research found that value-led and communal elements are in the center of the spatio-cultural process, as they build the foundation for tangible sustainability in the informal settlements.

A second significant finding is that the examination through the spatial framework also reveals elements of sustainability. Therefore, the overlapping of verbal and visual data led to the insight that both findings originated from the same spatio-cultural system and coincide perfectly with each other. They can be categorised under the same codes effortlessly (see Table 2). This also shows that the space of the informal settlements is imbued with cultural values and represents physical depictions of the intangible cultural heritage saved in memory, experience and the rural past. The analysis implies that culture can be measured and recorded in a spatial form, as the organic placemaking reflects the cultural heritage of the rural migrants in day-to-day life. Having the two-pronged approach generating similar sustainability features confirms the importance of informalities for Islamabad.

Figure 7 represents the process of cultural proliferation occurring in the spatial realm of informal communities. It highlights the need to propose a solution to prevent the loss of valuable heritage by reducing the effects of the constraints through proposition of policy targeting the five stated spatio-cultural sustainability elements. A multifaceted intervention to remove the constraints/barriers and facilitate the translation of intangible culture into the spatial realm of the informal settlements needs to be made. The conclusion drawn from the analysis using the two-pronged methodological approach is that, based on their cultural heritage, the informal dwellers can find solutions to most of the problems; however, they need a facilitation strategy to ease this process and curtail the constraints/barriers preventing them from implementing sustainability actions themselves. Experts and policy makers can play the role of facilitators who can ameliorate the effects of the constraints/barriers and assist the informal community to fully translate intangible cultural heritage to the spatial setting in which they dwell to re-create a tangible spatio-cultural heritage. This re-created tangible spatio-cultural heritage will be a medium to preserve the sustainable attributes found in the cultural legacy of the region; it will save local sustainability values and strategies from extinction and, in doing so, lead to sustainable placemaking from which the city can benefit.

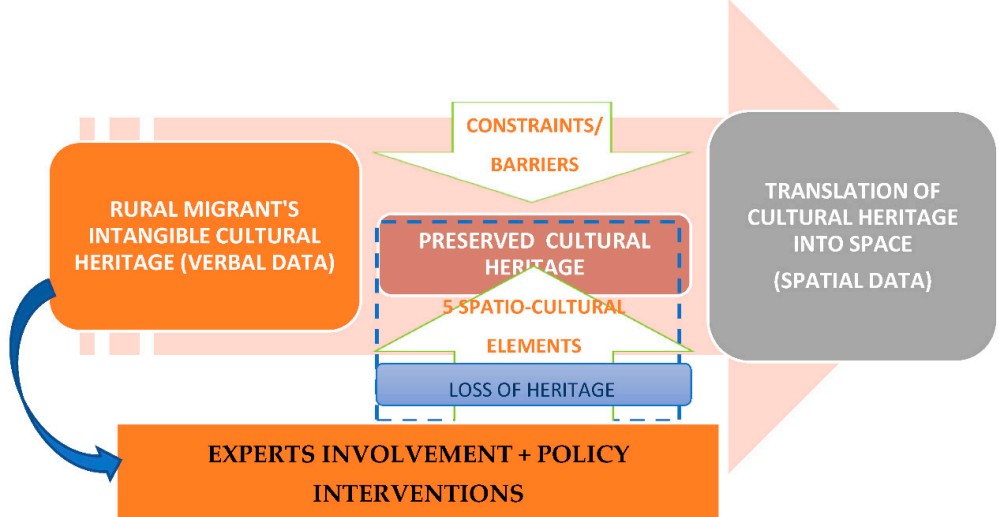

**Figure 7.** Spatio-cultural process of the relationship of culture with place.

## 5. Discussion, Methodical Limitations and Future Research

Informal settlements are increasingly attracting research attention as spaces of cultural and social dynamics with the potential to improve urban sustainability. Many studies apply mixed-methods to build up a better picture with practical implications. A study of an informality in Dar Es Salaam, Tanzania [46] (p. 459) similarly applied mapping and transect walks to explore the creation of collective spaces which maintain "the life experiences from rural areas with their common norms, social, traditional and cultural values" and contribute to urban sustainability. It argues that developers, urban planners and the city authorities should acknowledge the enriching activities and practices that informalities bring to the cities. The importance of streets as dynamic productive places where local economies and cultural and social activities thrive is emphasised in the comparative analysis of urban informalities across the world by Carracedo [47]. He concludes that any regeneration strategy should be based on understanding the original physical and spatial organization patterns in the informality which echoes our findings that such communities represent pockets of valuable cultural heritage. In a Sierra Leoni context, Rigon et al. [48] explain that the informalities provide to the burgeoning urban dwellers what the formal sector is unable to do, namely a livelihood and social protection, challenging urban inequalities and making cities more sustainable.

Methodologically, mixed methods involving grounded theory have been seen as a way "to reveal informal aspects and freshly emerging themes" when case studies are analysed in their complexity at various levels [49] (p. 935). Urban informalities also generate informal economic activities, which in the South American context have been studied using grounded theory to reveal the complexity of two types of logic used by local entrepreneurs, that of community and market, in order to establish a polycentric system of rules and norms of interaction [50]. We see similar ways of navigating the spatio-cultural elements in the two investigated Pakistani informal settlements in order to create livelihoods. Grounded theory provides the opportunity to uncover latent social aspects, provide fresh perspectives and help in analysing nuances in human behaviours and culture. This is particularly valuable where cultural barriers exist, such as in women's access to maternal health care in India's informal settlements [51], Muslim women's work in Oman's tourism industry [52], or preserving social capital in the slums of Zurabad in Iran [53]. In our case, the grounded theory approach facilitated access to and better understanding of social groups that are often ostracised within the context of the modern capital of Pakistan. Similar to the analysis conducted by Woodcraft et al. [54], grounded theory revealed real empirical insights about the capacities and capabilities of the local communities living in the urban informalities. This research adds to the body of knowledge about informalities

by applying an original methodology to geographical and anthropological areas whose importance has only recently started to be recognized.

The two-pronged mixed-methods methodological approach applied in our analysis delivered very similar outcomes while providing a deeper understanding of the Pakistani informalities. Grounded theory and spatial mapping generated verbal and visual data which described synergistically the same socio-cultural elements of the informalities. This may be partially due to the fact that the two methods applied do not strictly fall within the dichotomy of qualitative and qualitative. While grounded theory is purely qualitative, mapping incorporates both quantitative and qualitative aspects.

However, this may not always be the case. The mixed-methods approach was applied to only two case studies, and it would have been helpful if more sites were selected. As two informal settlements with distinctive characteristics were chosen within Islamabad, this may pose challenges for analytical generalization of the findings to other settings. Nevertheless, the learnings from this analysis and the other evidence from the literature can inform different situations, not only in terms of policy insights but also in relation to the application of a suite of methods that bring together the spatial and cultural dimensions of a city.

There are opportunities to further develop the ideas brought by this study by exploring the inhabitants' attitudes and values concerning the urban informalities and the lives they have left behind through in-depth ethnographies. The photographs in the study are used illustratively, but they can further be analyzed and interpreted as ethnographic data. Longer engagement with the informal settings through observational, visual and unstructured interviews could also be a goal for further ethnographic research. Further attention to the social and demographic diversity of the informal settlements can also be given in order to create a more detailed picture of the urban dwellers, including children, women and labourers who work outside the informalities.

## 6. Conclusions

The rural lifestyle needs to be appreciated for its socially interactive and ecologically-friendly attributes. These aesthetic, cultural and spiritual aspects that are kept alive in rural communities have been diminishing rather rapidly in urban environments. The findings from this study suggest that informal settlements provide a medium for rural heritage to be transferred to the cities and, hence, provide an opportunity to the planners and policy makers for cultural preservation and the improvement of the sustainable livability within the city.

An alternative approach towards dealing with urban informal settlements is put forward which argues for treating them as pockets of rural cultural heritage within the city, where local spatio-cultural processes can provide viable alternatives to the spatial planning framework governing the formally planned neighborhoods. The research negates the need for solutions based on theories developed in the Global North to promote sustainable development in the informal settlements of Islamabad or other cities of the Global South. Thus, the research endorses cultural specificity to ensure sustainability in the formal and informal communities of the city.

The overlapping of visual and verbal data collected through the two frameworks of spatial analysis and grounded theory helped in generating findings about the spatio-cultural process of informal settlements in a comprehensive manner. Grounded theory in particular created insights into the complex social aspects and human perspectives expressed by the residents of the urban informalities, which confirm the importance of these settlements. The spatial analysis identified the spaces and their uses valued by these communities. Foremost was the finding that the cultural values and social system of the informal community are based on the underlying concepts of sustainability. The in-depth interviews explored and revealed that sustainability is an evident notion in the cultural process. Furthermore, the findings from both methodologies described how cultural heritage can be visually and verbally documented by using a combination of data collection

tools during transect walks with group discussions, photography, observation and in-depth interviews. It was shown how mapping can effectively assemble data on cultural heritage, making a case for a methodological approach to extract intangible information concerning social and cultural values and to translate this into spatial maps. This methodology has been exploratory in nature, driven by the mixed-methods approach and can be used in applied research projects and for sustainable urban planning of informal settlements.

Using the two case studies in Islamabad, the research observed, documented and highlighted the rural sustainable livability which is absorbed into the informal settlements of the rapidly burgeoning cities of the developing world. These fast-paced and dynamic cities are constantly evolving, absorbing different trends followed across the globe, adapting to technological advancement and coping with the socio-political processes in the world [55]. The city is not rigid but fluid in nature, assimilating both tangible and intangible forces and molding itself accordingly. Regional cultural heritage has a contribution to make to city life in terms of sustainability through informal communities.

The research challenged the binary discourse of formal–informal settlements by exploring them through the spatio-cultural processes taking place in the case studies. It attempted to look beyond them as unclean, disorderly and illegitimate, and rather as places fabricated by the local people themselves [56]. The study explored the presence of valuable sustainability assets latent in the spatio-cultural system of informal settlements. It contends that cultural heritage entails sustainability and provides empirical evidence for this.

Informal settlements are presented as part of the solution rather than a problem for the urban realm [57]. It appears that the dilapidation of informal settlements is a result of constraints which hamper the process of cultural proliferation, leading to loss of heritage. The argument presented in the study states that interventions to ease the spatio-cultural process need to be proposed that help in removing the constraints and aid the cultural transfer into a place. Such interventions should take a bottom-up approach by working with the existing spatio-cultural assets of the community (documented through maps) and utilizing the potential assets of the community (discovered as codes and themes in NVivo). During the exploration of the tangible and intangible cultural assets of the community, constraints and barriers can also be identified. Experts can be involved as facilitators to make the spatio-cultural process possible. It is anticipated that this insight will result in the preservation of cultural heritage in informal settlements making the urban environment more sustainable.

**Author Contributions:** R.S. and D.M. conceptualised this study; R.S. collected and analysed the empirical data and drafted the article; both authors made contributions throughout all sections, read and approved the final manuscript for publication. All authors have read and agreed to the published version of the manuscript.

**Funding:** Ramisa Shafqat received an Australian Postgraduate Research Scholarship.

**Institutional Review Board Statement:** The study was approved by the Human Research Ethics Committee of Curtin University.

**Informed Consent Statement:** Informed consent was obtained from all subjects involved in the study.

**Data Availability Statement:** Further information regarding data is available upon request from the corresponding author.

**Acknowledgments:** Ramisa Shafqat acknowledges the Australian Postgraduate Research Scholarship and Curtin Publication Grant which helped in conducting and preparing this research for publication.

**Conflicts of Interest:** The authors declare no conflict of interest.

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
