# Peer review of "Using Mixed Methods to Understand Spatio-Cultural Process in the Informal Settlements: Case Studies from Islamabad, Pakistan"

_humans, doi:10.3390/humans2040017_

Round 1
Reviewer 1 Report
My comments are included in the attached file.

Author Response
Please refer to response report attached.

Author Response
Please find attached the response report

Round 2
Reviewer 2 Report
Substantive: two sections on grounded theory (3.2 and 4.2) are not well integrated in the paper: the abstract and discussion/conclusions don't mention grounded theory at all. Thus, the importance of these two sections is not clear to the reader. Are they essential to the article's argument about the value of mixed methods? If not, delete. If they are, then re-frame the article to provide a rationale for them.
Minor writing: revise lines 147-148 to read: Mixed methods have been used for the last 30 years or so...
Author Response
The comments made by the Reviewer are incorporated into the revised draft. Abstract, discussion and conclusion sections are tweaked as per the suggestions made by the reviewer.